# THE SYMMETRIC GENERALIZED EIGENVALUE PROBLEM AS A NASH EQUILIBRIUM

**Ian Gemp**[*]

DeepMind

London, UK

imgemp@deepmind.com

**Charlie Chen**[*]

DeepMind

London, UK

ccharlie@deepmind.com

**Brian McWilliams**[†]

Google Research

Zürich, Switzerland

bmcw@google.com

## ABSTRACT

The symmetric generalized eigenvalue problem (SGEP) is a fundamental concept in numerical linear algebra. It captures the solution of many classical machine learning problems such as canonical correlation analysis, independent components analysis, partial least squares, linear discriminant analysis, principal components and others. Despite this, most general solvers are prohibitively expensive when dealing with *streaming data sets* (i.e., minibatches) and research has instead concentrated on finding efficient solutions to specific problem instances. In this work, we develop a game-theoretic formulation of the top-$k$ SGEP whose Nash equilibrium is the set of generalized eigenvectors. We also present a parallelizable algorithm with guaranteed asymptotic convergence to the Nash. Current state-of-the-art methods require $\mathcal{O}(d^2 k)$ runtime complexity per iteration which is prohibitively expensive when the number of dimensions ($d$) is large. We show how to modify this parallel approach to achieve $\mathcal{O}(dk)$ runtime complexity. Empirically we demonstrate that this resulting algorithm is able to solve a variety of SGEP problem instances including a large-scale analysis of neural network activations.

## 1 INTRODUCTION

This work considers the symmetric generalized eigenvalue problem (SGEP),

$$Av = \lambda Bv \tag{1}$$

where $A$ is symmetric and $B$ is symmetric, positive definite. While the SGEP is not a common sight in modern machine learning literature, remarkably, it underlies several fundamental problems. Most obviously, when $A = X^\top X$, $B = I$, and $X$ is a data matrix, we recover the ubiquitous SVD/PCA. However, by considering other forms of $A$ and $B$ we recover other well known problems. In general, we assume $A$ and $B$ consist of sums or expectations over outerproducts (e.g., $X^\top Y$ or $\mathbb{E}[xy^\top]$) to enable efficient matrix-vector products. These include, but are not limited to:

**Canonical Correlation Analysis (CCA):** Given a dataset of *paired* observations (or views) $x \in \mathbb{R}^{d_x}$ and $y \in \mathbb{R}^{d_y}$ (e.g., gene expressions $x$ and medical imaging $y$ corresponding to the same patient), CCA returns the linear projections of $x$ and $y$ that are maximally correlated. CCA is particularly useful for learning multi-modal representations of data and in semi-supervised learning (McWilliams et al., 2013); it is effectively the multi-view generalization of PCA (Guo & Wu, 2019) where $A$ and $B$ contain the cross- and auto-covariances of the two views respectively:

$$A = \begin{bmatrix} \mathbf{0} & \mathbb{E}[xy^\top] \\ \mathbb{E}[yx^\top] & \mathbf{0} \end{bmatrix} \qquad\qquad B = \begin{bmatrix} \mathbb{E}[xx^\top] & \mathbf{0} \\ \mathbf{0} & \mathbb{E}[yy^\top] \end{bmatrix}. \tag{2}$$

---

[*]Asterisk denotes equal contribution.

[†]Work done while at DeepMind.

**Independent Component Analysis (ICA):** ICA seeks the directions in the data which are most structured, or alternatively, appear least Gaussian (Hyvärinen & Oja, 2000). A common SGEP formulation of ICA uncovers latent variables which maximize the non-Gaussianity of the data as defined by its excess kurtosis. ICA has famously been proposed as a solution to the so-called cocktail party source-separation problem in audio processing and has been used for denoising and more generally, the discovery of explanatory latent factors in data. Here $A$ and $B$ are the excess kurtosis and the covariance of the data respectively (Parra & Sajda, 2003):

$$A = \mathbb{E}[\langle x, x \rangle xx^\top] - tr(B)B - 2B^2 \qquad\qquad B = \mathbb{E}[xx^\top]. \qquad (3)$$

**Normalized Graph Laplacians:** The graph Laplacian matrix ($L$) is central to tasks such as spectral clustering ($A = L$, $B = I$) where its eigenvectors are known to solve a relaxation of min-cut (Von Luxburg, 2007). Alternatives, such as the random walk normalized Laplacian ($A = L$, $B$ is the diagonal node-degree matrix), approximate other min-cut objectives. These normalized variants, in particular, are important to computing representations for learning value functions in reinforcement learning such as successor features (Machado et al., 2017a; Stachenfeld et al., 2014; Machado et al., 2017b), an extension of proto-value functions (Mahadevan, 2005) which uses the un-normalized graph Laplacian ($A = L$, $B = I$).

Partial least squares (PLS) can be formualted similarly to CCA and finds extensive use in chemometrics (Boucher et al., 2015), medical domains (Altmann et al., 2021) and beyond (McWilliams & Montana, 2010). Likewise, linear discriminant analysis (LDA) can be formulated as a SGEP and learns a label-aware projection of the data that separates classes well (Rao, 1948). More examples and uses of the SGEP can be found in (Bie et al., 2005; Borga et al., 1997). We now shift focus to the mathematical properties and challenges of the corresponding SGEP.

In this work, we assume the matrices $A$ and $B$ above can either be defined using expectations under a data distribution (e.g., $\mathbb{E}_{x \sim p(x)}[xx^\top]$) or means over a finite sample dataset (e.g., $\frac{1}{n}X^\top X$ where $X \in \mathbb{R}^{n \times d_x}$). In either case, we typically assume the data has mean zero unless specified otherwise.

Note that the SGEP, $Av = \lambda Bv$, is similar to the eigenvalue problem $B^{-1}Av = \lambda v$. There are two reasons for working with the SGEP instead: 1) inverting $B$ is prohibitively expensive for a large matrix and 2) while $A$ and $B \succ 0$ are symmetric, $B^{-1}A$ is not, which hides useful information about the eigenvalues and eigenvectors (they are necessarily real and $B$-orthogonal). This also highlights that the SGEP is a fundamentally more challenging problem than SVD and why a direct application of previous game-theoretic approaches such as (Gemp et al., 2021; 2022) is not possible.

The complexity of solving the SGEP is $\mathcal{O}(d^3)$ where $d$ is the dimension of the square matrix $A$ (equiv. $B$). Several libraries exist for solving the SGEP in-memory (Tzounas et al., 2020). There is also a vast numerics literature we cannot do justice that considers large matrices (Sorensen, 2002).

We specifically focus on the stochastic, streaming data setting which is of particular interest to machine learning methods which learn by iterating over small minibatches of data (e.g., stochastic gradient descent). Under this setting, machine learning research has developed simple approximate solvers for singular value decomposition (SVD) that scale to very large datasets (Allen-Zhu & Li, 2017b). Similarly, in this work, we contribute a simple, elegant solution to the SGEP, including

- A game whose Nash equilibrium is the top-$k$ SGEP solution,
- An easily parallelizable algorithm with $\mathcal{O}(dk)$ per-iteration complexity relying only on matrix-vector products,
- An empirical analysis of neural similarity on activations $1000\times$ larger than prior work.

The game and accompanying algorithm are developed synergistically to achieve a formulation that is amenable to analysis and naturally leads to an elegant and efficient algorithm.

## 2 GENERALIZED EIGENGAME: PLAYERS, STRATEGIES, AND UTILITIES

In this work, we take the approach of defining the top-$k$ SGEP as a $k$-player game. It is an open question how to define a $k$-player game appropriately such that key properties of the SGEP are

captured[1]. As argued in previous work (Gemp et al., 2021; 2022), game formulations make obvious how computation can be distributed over players, leading to high parallelization, which is critical for processing large datasets. They have also clarified geometrical properties of the problem.

Specifically, we are interested in solving the top-$k$ SGEP which means we are interested in finding the (unit-norm) generalized eigenvectors $v_i$ associated with the top-$k$ largest generalized eigenvalues $\lambda_i$. Therefore, let there be $k$ **players** denoted $i \in \{1, \ldots, k\}$, and let each select a vector $\hat{v}_i$ (**strategy**) from the unit-sphere $\mathcal{S}^{d-1}$ (*strategy space*). We define player $i$'s **utility** function conditioned on its parents (players $j < i$) as follows:

$$u_i(\hat{v}_i | \hat{v}_{j<i}) = \overbrace{\frac{\langle \hat{v}_i, A\hat{v}_i \rangle}{\langle \hat{v}_i, B\hat{v}_i \rangle}}^{\substack{\text{generalized} \\ \text{Rayleigh Quotient}}} - \sum_{j<i} \frac{\langle \hat{v}_j, A\hat{v}_j \rangle \langle \hat{v}_i, B\hat{v}_j \rangle^2}{\langle \hat{v}_j, B\hat{v}_j \rangle^2 \langle \hat{v}_i, B\hat{v}_i \rangle} \tag{4}$$

$$= \underbrace{\hat{\lambda}_i}_{\text{reward}} - \sum_{j<i} \underbrace{\hat{\lambda}_j \langle \hat{y}_i, B\hat{y}_j \rangle^2}_{\text{penalty}} \quad \text{where } \hat{y}_i = \frac{\hat{v}_i}{||\hat{v}_i||_B}, \tag{5}$$

$\hat{\lambda}_i = \frac{\langle \hat{v}_i, A\hat{v}_i \rangle}{\langle \hat{v}_i, B\hat{v}_i \rangle}$, and $||z||_B = \sqrt{\langle z, Bz \rangle}$.

Player $i$'s utility has an intuitive explanation. The first term is recognized as the generalized Rayleigh quotient which can be derived by left multiplying both sides of the SGEP ($v^\top A v = \lambda v^\top B v$) and solving for $\lambda$. Note that the generalized eigenvectors are guaranteed to be $B$-orthogonal, i.e., $v_i^\top B v_j = 0$ for all $i \neq j$ (Appx. A Lemma 3). Therefore, the *reward* term incentivizes players to find directions that result in large eigenvalues, but are simultaneously *penalized* for choosing directions that align with their parents (players with index less than $i$, higher in the hierarchy). Finally, the penalty coefficient $\hat{\lambda}_j$ serves to balance the magnitude of the penalty terms with the reward term such that players have no incentive to "overlap" with parents. In Appx. B Proposition 4, we derive these same utilities via a *deflation* perspective. Next, we formally prove these utilities are well-posed in the sense that, given exact parents, their optima coincide with the top-$k$ SGEP solution.

**Lemma 1** (Well-posed Utilities). *Given exact parents and assuming the top-k eigenvalues of $B^{-1}A$ are distinct and positive, the maximizer of player $i$'s utility is the unique generalized eigenvector $v_i$ (up to sign, i.e., $-v_i$ is also valid).*

Note that $\hat{\lambda}_i = \frac{\langle \hat{v}_i, A\hat{v}_i \rangle}{\langle \hat{v}_i, B\hat{v}_i \rangle} = \frac{\langle \hat{v}_i / ||\hat{v}_i||_B, A\hat{v}_i / ||\hat{v}_i||_B \rangle}{\langle \hat{v}_i / ||\hat{v}_i||_B, B\hat{v}_i / ||\hat{v}_i||_B \rangle} = \frac{\langle \hat{y}_i, A\hat{y}_i \rangle}{\langle \hat{y}_i, B\hat{y}_i \rangle}$, therefore, the results above still hold for utilities defined using vectors constrained to the unit ellipsoid, $||\hat{y}_i||_B = 1$, rather than the unit-sphere, $||\hat{v}_i||_I = 1$. However, in our setting, $B$ is a massive matrix which can never be explicitly constructed and instead only observed via minibatches. It is then not clear how to handle the constraint $||\hat{y}_i||_B = 1$. We therefore only consider an approach that assumes $||\hat{v}_i||_I = 1$.

Next, we provide intuition for the shape of these utilities. Surprisingly, while non-concave, we prove analytically in Appx. B that they have a simple sinusoidal shape. A numerical illustration is given in Figure 1 to help the reader visualize this property.

**Proposition 1** (Utility Shape). *Each player's utility is periodic in the angular deviation (θ) along the sphere. Its shape is sinusoidal, but with its angular axis (θ) smoothly deformed as a function of B. Most importantly, every local maximum is a global maximum.*

Figure 1 illustrates a primary difficulty of solving the SGEP over SVD. Due to the extreme differences in curvature caused by the $B$ matrix, the SGEP should benefit from optimizers employing adaptive per-dimension learning rates. To our knowledge, this 1-d visualization of the difficulty of the SGEP is novel and we exploit this insight in experiments.

Finally, we formally define our proposed game and prove its equilibrium constitutes the top-$k$ SGEP solution. We use the Greek letter **gamma** to denote **generalized**, and we differentiate between the game and the algorithm with upper $\Gamma$ and lower case $\gamma$ respectively.

**Definition 1** (Γ-EigenGame). *Let Γ-EigenGame be the game with players $i \in \{1, \ldots, k\}$, their strategy spaces $\hat{v}_i \in \mathcal{S}^{d-1}$, and their utilities $u_i$ as defined in equation (48).*

---

[1]The Courant-Fischer min-max principle poses the $i$th generalized eigenvalue as the solution to a two-player, zero-sum game (Parlett, 1998)—see Appx. A.2 for further discussion.

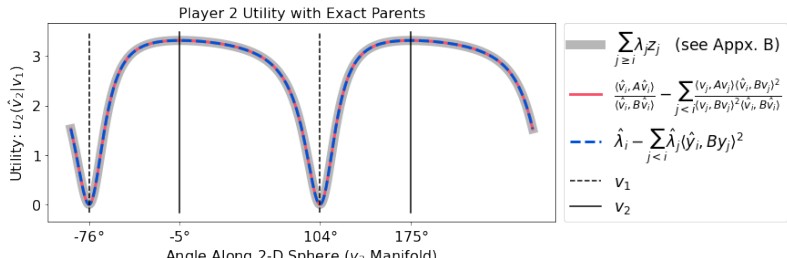

Figure 1: Each player's utility is a sinusoid on the sphere warped tangentially along the axis of angular deviation according to $B$; values for $A$ and $B$ used in this example are given in Appx. B. Three mathematical representations of the utility are plotted; their equivalence is supported by the overlapping curves. If player 2 aligns with the top eigenvector (dashed vertical), they receive zero utility. If they align with the second eigenvector (solid vertical), they receive $\lambda_2$ (optimal) as reward. If $B = I$ as in SVD/PCA, the vertical lines indicating the minima and maxima would be separated by exactly $90°$. In this case, the matrix $B$ redefines what it means for two vectors to be orthogonal ($\langle \hat{v}_i, B\hat{v}_j \rangle = 0$), so that the vectors are $71°$ (equivalently, $180° - 71° = 109°$) from each other.

**Theorem 1** (Nash Property). *Assuming the top-$k$ generalized eigenvalues of the generalized eigenvalue problem $Av = \lambda Bv$ are positive and distinct, their corresponding generalized eigenvectors form the unique, strict Nash equilibrium of $\Gamma$-EigenGame.*

*Proof.* Lemma 1 proves that each generalized eigenvector $v_i$ ($i \in \{1, \ldots, k\}$) is the unique best response to $v_{-i}$, which implies the entire set constitutes the unique Nash equilibrium. $\square$

## 3 ALGORITHM: UNBIASED PLAYER UPDATES AND AUXILIARY VARIABLES

Given that $\Gamma$-EigenGame suitably captures the top-$k$ SGEP, we now develop an iterative algorithm to approximate its solution. The basic approach we take is to perform parallel gradient ascent on all player utilities simultaneously. We focus on this approach in particular because it aligns with the predominant machine learning paradigm and hardware. We will first write down the gradient of each player's utility and then introduce several simplifications for the purpose of enabling unbiased estimates in the stochastic setting.

Up to scaling factors, the gradient of player $i$'s utility function with respect to $\hat{v}_i$ is

$$\frac{(\hat{v}_i^\top B \hat{v}_i) A \hat{v}_i - (\hat{v}_i^\top A \hat{v}_i) B \hat{v}_i}{\langle \hat{v}_i, B \hat{v}_i \rangle^2} - \sum_{j < i} \frac{\hat{\lambda}_j}{\langle \hat{v}_j, B \hat{v}_j \rangle} (\hat{v}_i^\top B \hat{v}_j) \frac{\left[ \langle \hat{v}_i, B \hat{v}_i \rangle B \hat{v}_j - \langle \hat{v}_i, B \hat{v}_j \rangle B \hat{v}_i \right]}{\langle \hat{v}_i, B \hat{v}_i \rangle^2}. \quad (6)$$

See Lemma 5 in Appx. B for a derivation of the gradient. Recall that $B$ is a matrix that we intend to estimate with samples, i.e., it is a random variable, and it appears several times in the denominator of the gradient. Obtaining unbiased estimates of inverses of random variables is difficult (e.g., the naive approach gives an overestimate; $\mathbb{E}[1/x] \geq 1/\mathbb{E}[x]$ by Jensen's inequality). We can remove the scalar $\langle \hat{v}_i, B \hat{v}_i \rangle^2$ in the denominator because it is common to all terms and will not change the direction of the gradient nor the location of fixed points; this step is critical to the design of our stochastic algorithm which we will explain later. We also use the following two additional relations:

(i) $\hat{\lambda}_j \langle \hat{v}_i, B \hat{v}_j \rangle = \langle \hat{v}_i, A \hat{v}_j \rangle$ if player $i$'s parents match their true solutions, i.e., $\hat{v}_{j<i} = v_{j<i}$,

(ii) $\sqrt{\langle \hat{v}_j, B \hat{v}_j \rangle} = ||\hat{v}_j||_B$ is strictly positive and real-valued because $B \succ 0$,

to arrive at the simplified update direction

$$\tilde{\nabla}_i = \overbrace{(\hat{v}_i^\top B \hat{v}_i) A \hat{v}_i - (\hat{v}_i^\top A \hat{v}_i) B \hat{v}_i}^{\text{reward}} - \sum_{j<i} \overbrace{(\hat{v}_i^\top A \hat{y}_j) \left[ \langle \hat{v}_i, B \hat{v}_i \rangle B \hat{y}_j - \langle \hat{v}_i, B \hat{y}_j \rangle B \hat{v}_i \right]}^{\text{penalty}}. \quad (7)$$

---

**Algorithm 1** Deterministic / Full-batch $\gamma$-EigenGame

---

1: Given: $A \in \mathbb{R}^{d \times d}$ and $B \in \mathbb{R}^{d \times d}$, step size sequence $\eta_t$, and number of iterations $T$.
2: $\hat{v}_i \sim \mathcal{S}^{d-1}$, i.e., $\hat{v}_i \sim \mathcal{N}(\mathbf{0}_d, \mathbf{I}_d)$; $\hat{v}_i \leftarrow \hat{v}_i / ||\hat{v}_i||$ for all $i$
3: **for** $t = 1 : T$ **do**
4:     **parfor** $i = 1 : k$ **do**
5:         $\hat{y}_j = \frac{\hat{v}_j}{\sqrt{\langle \hat{v}_j, B\hat{v}_j \rangle}}$
6:         $\texttt{rewards} \leftarrow (\hat{v}_i^\top B\hat{v}_i)A\hat{v}_i - (\hat{v}_i^\top A\hat{v}_i)B\hat{v}_i$
7:         $\texttt{penalties} \leftarrow \sum_{j<i}(\hat{v}_i^\top A\hat{y}_j)\big[\langle \hat{v}_i, B\hat{v}_i \rangle B\hat{y}_j - \langle \hat{v}_i, B\hat{y}_j \rangle B\hat{v}_i\big]$
8:         $\tilde{\nabla}_i \leftarrow \texttt{rewards} - \texttt{penalties}$
9:         $\hat{v}_i' \leftarrow \hat{v}_i + \eta_t \tilde{\nabla}_i$
10:        $\hat{v}_i \leftarrow \frac{\hat{v}_i'}{||\hat{v}_i'||}$
11:     **end parfor**
12: **end for**
13: return all $\hat{v}_i$

---

Simplifying the gradient using (1) is sound because the hierarchy of players ensures the parents will be learned exactly asymptotically. For instance, player 1's update has no penalty terms and so will converge asymptotically. The argument then proceeds by induction.

Note that $B$ still appears in the denominator via the $\hat{y}_j$ terms (recall equation (48)). We will revisit this issue later, but for now we will show this update converges to the desired solution given exact estimates of expectations (full-batch setting). Lemma 2 is a stepping stone to proving convergence with arbitrary parents in Theorem 2.

**Lemma 2.** *The direction $\tilde{\nabla}_i$ defined in equation (7) is a steepest ascent direction on utility $u_i(\hat{v}_i | \hat{v}_{j<i})$ given exact parents $\hat{v}_{j<i} = v_{j<i}$.*

*Proof.* This fact follows from the above argument that removing a positive scalar multiplier does not change the direction of the gradient of $u_i$ w.r.t. $\hat{v}_i$ and applying relation (i). $\square$

We present the deterministic version of $\gamma$-EigenGame in Algorithm 1 where $k$ players use $\tilde{\nabla}_i$ in (7) to maximize their utilities in parallel (see **parfor**-loop below). While simultaneous gradient ascent fails to converge to Nash equilibria in games in general, it succeeds in this case because the hierarchy we impose ensures each player has a unique best response (Lemma 1); this type of procedure is known as *iterative strict dominance* in the game theory literature. Theorem 2, proven in Appx. E, guarantees it converges asymptotically to the true solution.

**Theorem 2** (Deterministic / Full-batch Global Convergence). *Given a symmetric matrix $A$ and symmetric positive definite matrix $B$ where the top-$k$ eigengaps of $B^{-1}A$ are positive along with a square-summable, not summable step size sequence $\eta_t$ (e.g., $1/t$), Algorithm 1 converges to the top-$k$ generalized eigenvectors asymptotically ($\lim_{T \to \infty}$) with probability 1.*

In the big data setting, $A$ and $B$ are statistical estimates, i.e., expectations of quantities over large datasets. Precomputing exact estimates is computationally expensive, so we assume a data model that allows drawing small *minibatches* of data at a time. Under such a model, stochastic approximation theory typically guarantees that as long as the update directions are *unbiased*, i.e., equal in expectation to the updates with exact estimates, then an appropriate algorithm will converge to the true solution.

In order to construct an unbiased update direction given access to minibatches of data, we need to draw multiple minibatches independently at random. We can construct an unbiased estimate of products of expectations, e.g., $(\hat{v}_i^\top B\hat{v}_i)A\hat{v}_i$, by drawing an independent batch for each, e.g., one for $B$ and one for $A$. However, the $B$ that appears in the denominator of $\hat{y}_j$ is problematic; we cannot construct an unbiased estimate of the inverse of a random variable.

These problematic $\hat{y}_j$ terms only appear in the penalties, which are a function of the parents' eigenvector approximations. The first eigenvector has no parents, and so we can easily construct an unbiased estimate for it using multiple minibatches. We can then construct an unbiased estimate for

---

**Algorithm 2** Stochastic $\gamma$-EigenGame

---

1: Given: paired data streams $X_t \in \mathbb{R}^{b \times d_x}$ and $Y_t \in \mathbb{R}^{b \times d_y}$, number of parallel machines $M$ per player (minibatch size per machine $b' = \frac{b}{M}$), step size sequences $\eta_t$ and $\gamma_t$, scalar $\rho$ lower bounding $\sigma_{min}(B)$, and number of iterations $T$.

2: $\hat{v}_i \sim \mathcal{S}^{d-1}$, i.e., $\hat{v}_i \sim \mathcal{N}(\mathbf{0}_d, \mathbf{I}_d)$; $\hat{v}_i \leftarrow \hat{v}_i / ||\hat{v}_i|$ for all $i$

3: $[B\hat{v}]_i \leftarrow \hat{v}_i^0$ for all $i$

4: **for** $t = 1 : T$ **do**

5:    **parfor** $i = 1 : k$ **do**

6:       **parfor** $m = 1 : M$ **do**

7:          Construct $A_{tm}$ and $B_{tm}$ (*unbiased estimates using independent data batches)

8:          $\hat{y}_j = \dfrac{\hat{v}_j}{\sqrt{\max(\langle \hat{v}_j, [B\hat{v}]_j \rangle, \rho)}}$

9:          $[B\hat{y}]_j = \dfrac{[B\hat{v}]_j}{\sqrt{\max(\langle \hat{v}_j, [B\hat{v}]_j \rangle, \rho)}}$

10:          $\texttt{rewards} \leftarrow (\hat{v}_i^\top B_{tm} \hat{v}_i) A_{tm} \hat{v}_i - (\hat{v}_i^\top A_{tm} \hat{v}_i) B_{tm} \hat{v}_i$

11:          $\texttt{penalties} \leftarrow \sum_{j<i} (\hat{v}_i^\top A_{tm} \hat{y}_j) \big[ \langle \hat{v}_i, B_{tm} \hat{v}_i \rangle [B\hat{y}]_j - \langle \hat{v}_i, [B\hat{y}]_j \rangle B_{tm} \hat{v}_i \big]$

12:          $\tilde{\nabla}_{im} \leftarrow \texttt{rewards} - \texttt{penalties}$

13:          $\nabla_{im}^{Bv} = (B_{tm} \hat{v}_i - [B\hat{v}]_i)$

14:       **end parfor**

15:       $\tilde{\nabla}_i \leftarrow \frac{1}{M} \sum_m [\tilde{\nabla}_{im}]$

16:       $\hat{v}_i' \leftarrow \hat{v}_i + \eta_t \tilde{\nabla}_i$

17:       $\hat{v}_i \leftarrow \frac{\hat{v}_i'}{||\hat{v}_i'||}$

18:       $\nabla_i^{Bv} \leftarrow \frac{1}{M} \sum_m [\nabla_{im}^{Bv}]$

19:       $[B\hat{v}]_i \leftarrow [B\hat{v}]_i + \gamma_t \nabla_i^{Bv}$

20:    **end parfor**

21: **end for**

22: return all $\hat{v}_i$

---

each subsequent player by inductive reasoning. Intuitively, once the parents have been learned, $\hat{v}_j$ should be stable and so it should be possible to estimate $B\hat{v}_j$ from a running average, and in turn, $\hat{y}_j$. This suggests introducing an auxiliary variable, denoted $[B\hat{v}]_j$ to track the running averages of $B\hat{v}_j$ (a similar approach is employed in (Pfau et al., 2018)). This effectively replaces $B\hat{y}_j$ with a non-random variable, avoiding the bias dilemma, at the expense of doubling the number of variables. Note that introducing this auxiliary variable implies the inner product $\langle \hat{v}_j, [B\hat{v}]_j \rangle$ may not be positive definite, therefore, we manually clip the result to be greater than or equal to $\rho$, the minimum singular value of $B$.

Precise pseudocode is given in Algorithm 2. Differences to Algorithm 1 are highlighted in color (auxiliary differences in blue, clipping in red). We point out that introducing an auxiliary variable for player $i$ to track $[B\hat{v}]_i$ is not feasible because unlike player $i$'s parents' variables, $\hat{v}_i$ cannot be assumed non-stationary. This is why removing $\langle \hat{v}_i, B\hat{v}_i \rangle^2$ earlier from the denominator of equation (6) was critical. Lastly, note that these modifications to the update are derived using an understanding of the intended computation and theoretical considerations; put shortly, *autograd* libraries will not uncover this solution. See Appx. E.2 for more discussion and analysis of Algorithm 2.

**Computational Complexity and Parallelization.** The naive, per-iteration runtime and work costs of this update are $\mathcal{O}(bdk^2)$ with batch size $b$, but due to the simplicty of the update (i.e., purely matrix-vector products), there are several opportunities for both model and data parallelism to reduce runtime cost to $\mathcal{O}(dk)$ (see Appx. I for steps). Note that low aggregate batch sizes can induce high gradient variance, slowing convergence (Durmus et al., 2021), making data parallelism desireable.

To give a concrete example, if each player (model) parallelizes over $M = b$ machines (data) as indicated by the two **parfor**-loops, the complexity reduces to $\mathcal{O}(dk)$. This is easy to implement with modern libraries, e.g., pmap using Jax, and as in prior work (Gemp et al., 2021), the communication of parents $v_{j<i}$ between machines is efficient in systems with fast interconnects (e.g., TPUs) although this presents a bottleneck we hope to alleviate in future work. Alternative parallel implementations are discussed in Appx. F. Lastly, the update consists purely of inexpensive

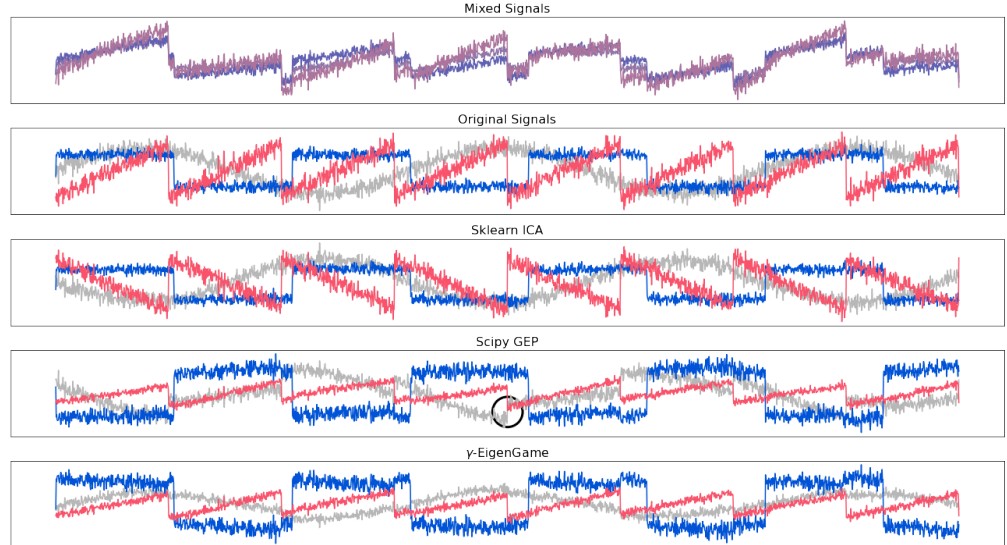

Figure 2: Blind Source Separation. Algorithm 2 ($\gamma$-EigenGame) run for 1000 epochs with mini-batches of size $\frac{n}{4}$ subsampled i.i.d. from the dataset recovers the three original signals from the linearly mixed signals. Scikit-learn's FastICA[2] also recovers the signals by maximizing an alternative measure of non-Gaussianity. Directly solving the SGEP using `scipy.linalg.eigh(A, B)` fails to cleanly recover the gray sinusoid (black circle highlights a discontinuity) due to overfitting to the sample dataset. We confirm this by training $\gamma$-EigenGame for many more iterations and show it exhibits similar artifacts in Appx. G.

elementwise operations and matrix-vector products that can be computed quickly on deep learning hardware (e.g., GPUs and TPUs); unlike previous state-of-the-art in (Meng et al., 2021), no calls to CPU-bound linear algebra subroutines are necessary.

## 4 RELATED WORK

The SGEP is a fundamental problem in numerical linear algebra with numerous applications in machine learning and statistics. There is a long history in numerical computing of solving large SGEP problems (Sorensen, 2002; Knyazev & Skorokhodov, 1994; Golub & Ye, 2002; Aliaga et al., 2012; Klinvex et al., 2013). Many methods iterate with what can be viewed as "gradient-like" updates (D'yakonov & Knyazev, 1982; D'yakonov & Knyazev, 1992), however, they are not immediately applicable in the stochastic, streaming data setting. To our knowledge, efficient approaches for the SGEP or specific sub-problems (e.g., CCA) scale at best $\mathcal{O}(d^2k)$ in the streaming data setting.

Ge et al. (2016) give an algorithm for top-$k$ SGEP that makes repeated use of a linear system solver to approximate the subspace of the true generalized eigenvectors, but may return an arbitrary rotation of the solution. While their method is theoretically efficient, it requires precomputing $A$ and $B$ which prohibits its use in a streaming data setting. The sequential least squares CCA algorithm proposed by Wang et al. (2016) similarly requires access to the full dataset up front, however, in their case, it is to ensure the generalized eigenvectors are exactly unit-norm relative to the matrix $B$. Allen-Zhu & Li (2017a) develop a SGEP algorithm that is theoretically linear in the size of the input ($nd$) and $k$, however, they assume access to the entire dataset (non-streaming). LOBPCG (Knyazev, 2017) is a non-streaming, Rayleigh maximizing technique with line-search and a preconditioned gradient.

Arora et al. (2017) propose a convex relaxation of the CCA problem along with a streaming algorithm with convergence guarantees. However, instead of learning $V_x \in \mathbb{R}^{d_x}$ and $V_y \in \mathbb{R}^{d_y}$ directly, it learns $M = V_x V_y^\top \in \mathbb{R}^{d_x \times d_y}$ which is prohibitively expensive to store in memory for high-dimensional problems. Moreover, the complexity of this algorithm is $\mathcal{O}(d^3)$ due to an expensive projection step requiring an SVD of $M$. They propose an alternative version *without* guarantees that reduces the cost per iteration to $\mathcal{O}(dk^2)$.

---

[2]Run with `logcosh` approximation to negentropy (see (Hyvärinen & Oja, 2000) for explanation).

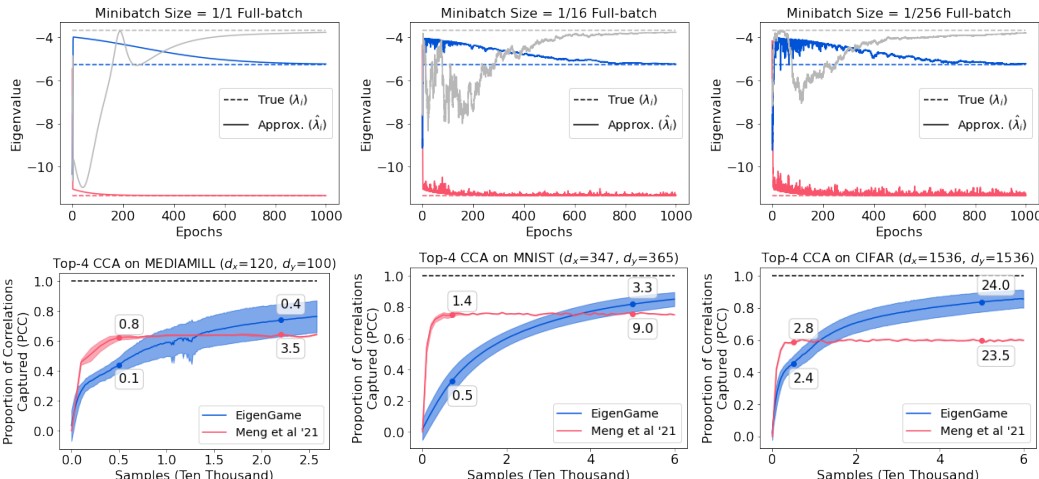

Figure 3: **Top:** $\gamma$-EigenGame converges to the true SGEP solution regardless of minibatch size in support of the unbiased nature of the derived update scheme (Algorithm 2). See Appx. G.2 for additional experiments showing learning in sequence performs worse than its parallel counterpart above. **Bottom:** $\gamma$-EigenGame compared to (Meng et al., 2021) on *proportion of correlations captured*—$\sum_i^k \hat{\lambda}_i / \sum_i^k \lambda_i$. Shading indicates $\pm 1$ stdev. Markers indicate runtime in seconds.

Gao et al. (2019) also consider the streaming setting, but instead focus on top-1 CCA and like (Wang et al., 2016) and (Allen-Zhu & Li, 2017a), use shift-invert preconditioning to accelerate convergence. Bhatia et al. (2018) solves top-1 SGEP in the streaming setting.

Most recently, Meng et al. (2021) proposed a method to estimate top-$k$ CCA in a streaming setting. Their algorithm requires several expensive Riemannian optimization subroutines, giving a per iteration complexity of $\mathcal{O}(d^2 k)$. Their convergence guarantee is in terms of subspace error, so as mentioned above, the projection matrices $V_x$ and $V_y$ may be rotations of their ordered (by correlation) counterparts. Their approach is the current state-of-the-art when considering CCA in the streaming setting for large datasets.

## 5 EXPERIMENTS

We demonstrate our proposed stochastic approach, Algorithm 2, on solving ICA and CCA via their SGEP formulations, and provide empirical support for its veracity. A Jax implementation is available at `github.com/deepmind/eigengame`. Scipy's `linalg.eigh(A, B)`(Virtanen et al., 2020) is treated as ground truth when the data size permits. Hyperparameters are listed in Appx. H.

**Independent Components Analysis.** ICA can be used to disentangle mixed signals such as in the cocktail party problem. Here, we use the SGEP formulation to unmix three linearly mixed signals. Note that because the SGEP learns a linear unmixing of the data, the magnitude (and sign) of the original signals cannot be learned. Any change in the magnitude of a signal extracted by the SGEP can be offset by adjusting the magnitude and sign of a mixing weight.

We replicate a synthetic experiment from `scikit-learn`(Pedregosa et al., 2011) and compare Algorithm 2 to several approaches. Figure 2 shows our stochastic approach ($\gamma$-EigenGame) is able to recover the shapes of the original signals (length $n = 2000$ time series).

**Implicit Regularization via Fixed Step Size Updates.** Note that if we run Algorithm 2 for $100\times$ more iterations with $1/10$th the step size, we converge to the exact SGEP solution (as found by `scipy`) and see similar artifacts in the extracted signals due to overfitting. Recently, Durmus et al. (2021) proved that fixed step size Riemannian approximation schemes converge to a stationary distribution around their solutions, which suggests $\gamma$-EigenGame enjoys a natural regularization property and explains its high performance on this the unmixing task. In Appx. G, we show that it is difficult to achieve similar results with `scipy` by regularizing $A$ or $B$ directly (e.g., $A + \epsilon I$) prior to calling `scipy.linalg.eigh`.

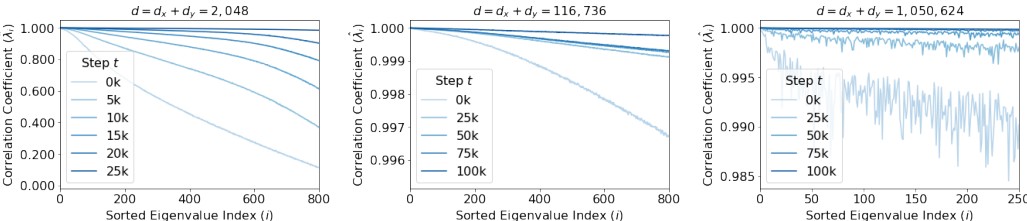

Figure 4: $\gamma$-EigenGame compares the representation (activations) of a deep network trained on CIFAR-10 for $t$ steps to that of its final learned representation ($25k$ or $100k$ steps). Curves with higher correlation coefficients indicate more similar representations.

**Unbiased Updates.** We empirically support our claim that the fixed point of Algorithm 2 is un-biased regardless of minibatch size.[3] Not only does $\gamma$-EigenGame recover the same generalized eigenvalues, but the top row of figure 3 also suggests that the algorithm takes a similar trajectory for each minibatch size.

**Canonical Correlations Analysis.** Here, we use $\gamma$-EigenGame to linearly project multimodal datasets into lower-dimensional spaces such that they are maximally correlated. As discussed in related work, several approaches have been developed to extend CCA to streaming, high-dimensional datasets. Recall that our approach has per-iteration complexity $\mathcal{O}(bdk)$ with the previous state-of-the-art in the streaming setting having $\mathcal{O}(d^2k)$ (Meng et al., 2021). We replicate the experiments of (Meng et al., 2021) and compare against their approach on three datasets.

The bottom row of figure 3 shows our approach is competitive with (Meng et al., 2021). We also point out that while the previous approach by Meng et al. (2021) enjoys theoretical convergence guarantees with rates, it appears to slow in progress near a biased solution. These datasets are low dimensional ($d \leq 3072$), so we are able to obtain ground truth eigenvectors efficiently using `scipy`. Our next set of experiments considers much higher dimensional where a naive call to `scipy` fails.

**Large-scale Neural Network Analysis.** Recently, CCA has been used to aid in interpreting the representations of deep neural networks (Raghu et al., 2017; Morcos et al., 2018; Kornblith et al., 2019). These approaches are restricted to layer-wise comparisons of representations, reduced-dimensionality views of representations (via PCA), or small dataset sizes to accomodate current limits of CCA approaches. We replicate one of their analyses (specifically Fig. 1a of (Morcos et al., 2018)) on the activations of an entire network (not just a layer), unblocking this type of analysis for larger deep learning models.

The largest dimensions handled in (Raghu et al., 2017) are $\mathcal{O}(10^3)$. Figure 4 demonstrates our approach (parallelized over 8 TPU chips) on $\mathcal{O}(10^3)$ dimensions (left), $\mathcal{O}(10^5)$ dimensions (middle), and $\mathcal{O}(10^6)$ dimensions (right). Note that in these experiments, we are loading minibatches of CIFAR-10 images, running them through a deep convolutional network, harvesting the activations, and then passing them to our distributed $\gamma$-EigenGame solver. As mentioned in Section 2, our understanding of the geometry of the utilities suggests replacing the standard gradient ascent on $\hat{v}_i$ with Adam (Kingma & Ba, 2014); Adam exhibits behavior that implicitly improves stability around equilibria (Gemp & McWilliams, 2019). For the smaller $\mathcal{O}(10^3)$ setting, where we can exactly compute ground truth using `scipy`, we confirm that our approach converges to the top-1024 (out of 2048 possible) eigenvectors with a subspace error of $0.002$ (see Appx. A).

## 6 CONCLUSION

We presented $\Gamma$-EigenGame, a game-theoretic formulation of the generalized eigenvalue problem (SGEP). Our formulation enabled the development of a novel algorithm that scales to massive streaming datasets. The SGEP underlies many classical data processing tools across the sciences, and we believe our proposed approach unblocks its use on the ever-growing size of modern datasets in the streaming setting. In particular, it achieves this by parallelizing computation using modern AI-centric distributed compute infrastructure such as GPUs and TPUs.

---

[3]The gray line converges last because we chose to minimize rather than maximize kurtosis.

**Acknowledgements.** We thank Claire Vernade for reviewing the paper and proofs and early discussions. Thore Graepel for early discussions. Zhe Wang for guidance on engineering. Sukhdeep Singh for project management.

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
