# OpenReview forum: "The Symmetric Generalized Eigenvalue Problem as a Nash Equilibrium"
_ICLR.cc/2023/Conference — ICLR 2023 notable top 25%_

### Official Review · Reviewer_4TCc · 2022-10-24

**Confidence:** 3
**Correctness:** 4
**Technical Novelty And Significance:** 4
**Empirical Novelty And Significance:** 4
**Recommendation:** 8

**Clarity, Quality, Novelty And Reproducibility:**


Other than the issues raised above, I do not have any other comments.

**Strength And Weaknesses:**

I think the writing in the paper could use some more work. Especially since most of the claims have short and simple proofs, it would be nice to include some more intuition about them in the main text. I verified most of the proofs in the supplementary materials. Some other concerns I have:
1. Theorem 1 (which follows from Lemma 1) does not require Proposition 1. And therefore Proposition 1 should not be before Definition 1 and Theorem 1. In fact, its only ever used in showing that Algorithm 2's updates are unbiased.

2. Lemma 1: $w_p$ are not defined. It would be nice to say they are arbitrary real coefficients and give some exposition on your end goal. For example, we now consider $\hat v_i$ as arbitrary linear combination of $v_i$'s. We will show that maximizing the utility for player is same as solving the following linear program: ...

3. Proposition 1: It would be nice to give a more definite reference for EigenGame (and results implying the utility function is cosine).


I think the claims about improving the per-iteration complexity should be followed by explanation about its implication on total time complexity. I think it should be noted that this algorithm does not have convergence rates and in fact, the error from parents on the time-complexity would show up in the number of iterations needed for the algorithm to converge. This is real and looking through proof of Theorem 2, it seems like the number of iterations even in the deterministic case scale exponentially in $k$ (since the $\epsilon$ error from parents lead to $\sqrt{\epsilon}$ error in children's fixed point).

That said, I would like to emphasize that the above comment does not dilute the contribution of this work.
I think the paper makes a good technical contribution to an important problem.

**Summary Of The Paper:**

This paper is concerned with solving the symmetric generalized eigenvalue problem $Av = \lambda B v$ , which generalizes various other well known problems like PCS, CCA, ICA etc. A particular approach chosen is to consider a corresponding game between k players and the rewards are set such that the Nash equilibrium is the top-k solution. What we already know about this problem: If we can invert B in the equation above, then SGEP can be solved as an SVD problem. This however loses structure (for example A and B are symmetric in SGEP) and therefore previous game theoretic approaches to solving SVD do not apply here.

Like mentioned above, the approach in this work is to construct a game between k players each selecting a strategy $v_i$. Their first main contribution is in constructing a sound utility function such that the Nash equilibrium of this game is the top-k SGEP solution. Intuitively, the utility for player i conditioned on players j < i (henceforth parents of i) is designed to: (1) incentivize finding directions that have large eigenvalues (called reward) (2) but simultaneously choosing directions which are far from directions chosen by the parents.

 To show that these rewards are sound, the work first shows that assuming $B^{-1}A$ has nice spectrum, and the parents converge to correct eigenvectors (called exact parents henceforth), the unique generalized eigenvector $v_i$ maximizes the player i's utility. This follows from noting that for the exact parents, the penalty term linearizes after a re-parameterization (and the reward term is already linear in the said re-parameterization). This shows that the top-k eigenvectors form the strict Nash equilibrium of this game.

Another particular important aspect of this utility design is that every local maximum is a global maximum here which follows from previous work. Together with above result, this suggests a simple gradient based algorithm. The work next shows under deterministic setting (A and B are known), this simple gradient based algorithm asymptotically converges. This requires proving that the error in eigenvectors for parents lead to bounded error in the solutions for the children. This result borrows ideas from previous work [Gemp et al. 2021].

Some care is needed to extend this to stochastic/streaming setting, we need the terms appearing in the denominator to be nicely conditioned and have to be handled appropriately. This algorithm has a per iteration complexity $dk$ which improves over the previous state-of-the-art per iteration complexity $d^2k$. This is followed by extensive experiments on CCA and ICA showing in practice this method is competitive.

**Summary Of The Review:**

I think this is a good contribution to understanding game theoretic version of various problems like ICA, PCA, CCA etc. Moreover, this work provides a very practical algorithm.

---

> ### Author Response · Authors · 2022-11-10
> **[4TCc] Response 1**
>
> Thank you for your comments, suggestions, and for reading the proofs!
>
> *Proposition 1 Placement*: Sorry for the confusion. Proposition 1 is meant to help the reader visualize the utility functions which we believe is helpful for building an intuition for both Theorem 1 and 2. Proposition 1 (along with Figure 1) provides intuition for how the global maxima of each utility function coincides with the true eigenvector direction. This supports Theorem 1 by showing the utilities are properly defined which is necessary for the Nash proof. Proposition 1 (along with Figure 1) also supports Theorem 2 by showing how every local maximum is also a global maximum, and therefore, simple gradient ascent will maximize this function despite it being non-convex on the manifold.
>
> *Appx B, Lemma 1*: Thank you for raising this and also for reading the appendix :). We have defined them and explained their purpose per your suggestion.
>
> *Cosine Utilities*: Absolutely. Thank you for pointing this out. We have added a more specific reference to Proposition 1 in Appx B (original EigenGame paper, Appx J.2).
>
> *Per-Iteration vs Total Time Complexity*: Good point. Reviewer U98f also mentioned this. Lowering the batch size increases the variance of the gradient estimates, which increases the number of iterations required to reach a given error tolerance. This type of effect is formalized in work such as Durmus et al [1] (e.g., Corollary 9 with variance defined in Assumption MD 1 on p. 3). We have added a citation to the section on complexity.
> One reason we emphasize the $\mathcal{O}(dk)$ result as a contribution is that some data is so high dimensional that only a few samples will fit on a device at a time. Having the ability to reduce the batch size (in the extreme case to 1), and then for example, employ variance reduction techniques (e.g., control variates, SVRG, …) makes the problem tractable; although, as you astutely point out, convergence can become a challenge.
>
> [1] On Riemannian Stochastic Approximation Schemes with Fixed Step-Size. Durmus et al. AISTATS '21.

---

### Official Review · Reviewer_dLpL · 2022-10-24

**Confidence:** 4
**Clarity, Quality, Novelty And Reproducibility:** All good.
**Correctness:** 4
**Technical Novelty And Significance:** 3
**Empirical Novelty And Significance:** 3
**Recommendation:** 6

**Strength And Weaknesses:**

I tried my best to check the proofs and they appear correct.

Strengths:
-) The paper introduces a nice setting/idea for eigenvalue computations.
-) Lots of intuition is presented and discussed.
-) Results demonstrate that the proposed method is effective.

-) The difference between eigengame-PCA and the current work is not properly explained.
-) The authors should discuss in more depth other optimization-based eigenvalue solvers such as TraceMIN or LOBPCG.
-) It's likely I missed something here, but why do the authors need to include the "Nash's equilibrium" phrase throughout their paper?
It seems to me that the algorithm developed is nothing more than an optimization-based eigenvalue solver for streamed datasets; which is absolutely fine for a publication. What am I missing?
-) The discussion about computational complexity is a bit confusing; why is $O(dk)$ after we parallelize across M=b batches? Is the cost per iteration of for-k loop only $O(d)$? I thought it would $O(dk)$ (which means the paralleized complexity is $O(dk^2)$). I must be missing something here.
-) My main concern is the following: are there really situations where I can not access the matrices A and B in a sequential batched manner? Because in this case the authors can use standard numerical linear algebra optimization-based approaches (i.e., out-of-core computing). In other words, in what applications $A$ and $B$ are accessed truly randomly?

**Summary Of The Paper:**

This paper presents an algorithm to compute the $k$ dominant eigenpairs of a streamed matrix. The algorithm is based on an optimization viewpoint where each individual eigenpair is maximizing a separate cost function. Numerical experiments verify the effectiveness of the proposed scheme.

**Summary Of The Review:**

The proposed method is interesting and the paper is well-written. My main concerns are: a) similarity with the PCA case, and b) realistic application in modern problems.

---

> ### Author Response · Authors · 2022-11-10
> **[dLpL] Response 1**
>
> Thank you for your comments and your pointers to related work.
>
> *EigenGame PCA vs EigenGame SGEP*: Thank you for raising this point of confusion. There are two paragraphs in the intro (1st and 7th) where we address the differences. We would ask that you please re-read these and let us know if this doesn’t help clarify things. The first instance reads “Most obviously, when $A=X^\top X$ , $B=I$, and $X$ is a data matrix, we recover the ubiquitous SVD/PCA", and the second instance ends with "this also highlights that the SGEP is a fundamentally more challenging problem than SVD". We also show how, in the specific case where $B=I$, EigenGame Unloaded (Gemp et al ‘22) can be derived from a reduction of our approach (Appx B, Proposition 3).
>
> *Other Approaches (TraceMIN and LOBPCG)*: Thank you for pointing these both out. In short, TraceMIN cannot be efficiently applied to the stochastic setting, but LOBPCG provokes some interesting ideas for future work.
> - TraceMIN [1] is an iterative approach that performs a series of operations at every step: 1) B-orthonormalization, 2) eigendecomposition of a $k \times k$ matrix, 3) the approximate solution of a saddle point problem, among others. The first step of $B$-orthonormalization precludes TraceMIN from being useful in the streaming data setting. This is because in order to $B$-orthonormalize a set of vectors, you need to know $B$. The only way to know $B$ is to first observe the entire data stream. Step 2 is also problematic if $k$ is large. The step of solving a saddle point problem approximately in the inner loop is still a large $d+k$ dimensional problem and would require a difficult “two time-scale” analysis as well.
> - LOBPCG is essentially power iteration plus line-search using a preconditioned gradient. Most approximate top-$k$ eigensolvers use some form of power iteration (as does ours). It would be interesting to consider incorporating preconditioning or line-search methods into our work. Both of these techniques have been explored in the Riemannian manifold setting [2, 3, 4].
>
> *Why Nash equilibrium?* Thank you for your question. You are missing that the formulation we pose is not an optimization problem. We have $k$ players that are simultaneously optimizing their own utility functions. This makes it a game. The canonical solution concept for a game is Nash equilibrium. If there was only one utility function, then it would be an optimization problem.
> It was already proven in prior work (Gemp et al ‘21) that no single objective function exists whose gradient comprises their update rule for all $k$ players. By reduction from our formulation to theirs in the specific setting of $B=I$, we also have that same result.
>
> *Computational Complexity*: Note we say "if **each** player parallalizes over $M = b$ machines". That means $k \times b$ machines total. Maybe you missed this difference? We draw out more of the steps in deriving the complexity at the very end of the appendix (Appx I Runtime Complexity). Please let us know if that does not clarify things.
>
> *Main Concern-- Assumptions on Reading $A$ and $B$*: In this work, we assume $A$ and $B$ are defined with a specific form via sums of outer products of high dimensional vectors, e.g., $A = \sum_i x_i x_i^\top$ and $B=I$ for SVD/PCA (see second to last sentence of 1st paragraph in intro). To our best knowledge, out-of-core computing typically assumes rows/columns of $A$ and $B$ can be accessed easily. However, notice that it is not easy (or efficient) to access a row/column of $A$ if it is defined via a sum of outer products because the $j$th row of $A$ is still defined via a sum: $A_j = \sum_i x_{ij} x_i$ where $x_{ij}$ is the $j$th entry of sample $x_i$. “Grabbing” the $j$th row then requires iterating over the entire dataset. We spell out several settings where $A$ and $B$ are defined via sums over outer products. Although this is clearly not true of all numerical eigenvalue problems, we argue that this is the predominant problem setting in machine learning.
> Furthermore, in probabilistic and statistical machine learning, datasets are, in fact, defined as distributions. Batches of data are considered samples from a true underlying distribution. Note that our algorithm is applicable in this setting as well.
>
> [1] Parallel implementations of the trace minimization scheme TraceMIN for the sparse symmetric eigenvalue problem. Klinvex et al. Elsevier - Computers \& Mathematics with Applications '13.
>
> [2] Newton acceleration on manifolds identified by proximal gradient methods. Bareilles et al. Springer - Mathematical Programming '22.
>
> [3] Riemannian stochastic quasi-Newton algorithm with variance reduction and its convergence analysis. Kasai et al. AISTATS '18.
>
> [4] Line search algorithms for locally Lipschitz functions on Riemannian manifolds. Hosseini et al. SIAM '18.

---

### Official Review · Reviewer_U98f · 2022-10-25

**Confidence:** 3
**Correctness:** 4
**Technical Novelty And Significance:** 3
**Empirical Novelty And Significance:** 3
**Recommendation:** 8

**Clarity, Quality, Novelty And Reproducibility:**

Regarding clarity, I refer the authors to points a) and b) above. Beyond the two points above, both the quality and the clarity of the writing looks good to me.

Regarding novelty, although the skeleton of the analysis and the main tools (e.g. penalty terms, error propagation etc.) are the same with the prior work for the standard (non-generalized) eigenvalue problem, the generalized eigenvalue problem poses unique challenges (providing unbiased gradient estimators) that this work addresses, at least partially on the theoretical front. On the empirical side, the proposed approach is shown to outperform the previous state of the art approaches.

Regarding reproducibility, the authors detail the hyper-parameters used in their experiments. They also plan to open source their code, making their experimental claims fully auditable by peers.


**Strength And Weaknesses:**

This work has three major strengths. First, the problem of generalized eigenvalue problem and its importance to the machine learning community is thoroughly motivated. Second, although there have been several works studying game-theoretic solutions to the standard (non-generalized) eigenvalue problem, these approaches are not directly applicable since the generalized eigenvalue problem is not equivalent to a standard eigenvalue problem for a symmetric positive definite $A$. Third, a direct generalization of previous approaches (for example using Equation (4) or (5)) poses unique challenges when applied to the stochastic (mini-batch) setting. Up to the technical issues outlined in Appendix E.2, the proposed approach resolves these challenges.

Regarding weaknesses, there are some points in this work that could benefit from additional elaboration.

a) $O(dk)$ iteration complexity property
The way I understand it, every algorithm that uses matrix vector products and has as many machines per player as the datapoints in the batch of $A$ and $B$ also avoids the $O(kd^2)$ factor in their per-iteration analysis. Effectively, it seems to me that the key claim is that the proposed algorithm only uses matrix vector products whereas prior art does not. I think clarifying if this is really the case and why prior work cannot adopt the same matrix-vector product interface would help.

Secondarily, as stated the result has some limitations. It is unclear how setting the batch size to be equal to the machine count per player affects the number of iterations required. It is not clear if we are trading off faster iterations and higher iteration counts.

b) Comparison with top-1 generalized eigenvector + deflation
This theoretical analysis of this work a lot of times argues that given (approximate) convergence of players $i < j$, player $j$ can approximately find the $j$th generalized eigenvector. To me this seems very similar to how we can use power iteration and variants to find the top eigenvector, apply a deflation step and then find the next eigenvector.

It is not clear to me if one could apply an algorithm for a top-1 generalized eigenvector like Bhatia et al. (2018) and with an appropriate deflation step find the next one and so forth. Is this actually the case? If so what are the limitations? If not, why?

A key advantage I can see is that here all players work concurrently whereas the other approach would find the eigenvectors sequentially. The theoretical analysis however does not seem to take advantage of that. Maybe an experimental comparison between updating all players concurrently and only updating after the convergence of previous players would help clarify this point to the reader.



**Summary Of The Paper:**

This work provides an algorithm for identifying the top-$k$ generalized eigenvalues for a pair of symmetric positive definite matrices $A$ and $B$. The algorithm works by reducing this problem to finding a Nash equilibrium to a multi-player game. For this game the authors prove that the gradient ascent algorithm provably converges asymptotically to the top-$k$ generalized eigenvalues.

The authors also seek to extend their algorithm to the stochastic setting where matrices $A$ and $B$ are accessible through stochastic estimates of their vector products via batching. They propose several modifications to their original objective that addresses several problems that arise with providing unbiased gradient estimates. While their modifications lead to unbiased estimators, they also make a complete and formal analysis harder and thus they leave it for future work. Empirically though, they do observe convergence on problems 1000x larger than the ones studied by prior work.

The ability to have independent machines processing independent mini-batches of $A$ and $B$, allows for low per-iteration complexity $O(dk)$ as compared to prior state of the art which requires $O(d^2k)$. The key is that matrix vector products cost $O(d)$ instead of $O(d^2)$ when considering minibatches of $O(1)$ samples.

**Summary Of The Review:**

I recommend this paper for acceptance. I am inclined to raise my score higher if the authors improve on the points raised in the "Strengths and Weaknesses" section.

---

> ### Author Response · Authors · 2022-11-10
> **[U98f] Response 1a**
>
> Thank you for your review and your concise summary. You are spot on.
>
> *Strengths*: Just to be clear so we’re on the same page, equations 4 and 5 are contributions of this paper. They were not introduced in prior work, which your comment seems like it might suggest.
>
> Weaknesses:
> - *$\mathcal{O}(dk)$ iteration complexity property*: Thank you for raising this. Your intuition is correct. The primary obstacle we most often see in prior work to achieving updates with pure matrix-vector products is the modeling of $B$-orthogonality as a hard constraint. This view requires all eigenvectors to be $B$-orthogonal at every iteration which precludes any approach that views $B$ via a minibatch stream. Incorporating $B$-orthogonality as a penalty into the SGEP problem formulation is what allows us to circumvent this issue. Any update composed purely of $\mathcal{O}(k)$ matrix-vector products can achieve $\mathcal{O}(dk)$ runtime complexity per iteration as you say. We have added a remark to this effect in the “Computational Complexity and Parallelization” section.
> - *Setting the batch size to 1* -- Thank you for your question. You are correct that reducing the batch size affects the convergence rate. Reviewer 4TCc also commented on this. Lowering the batch size increases the variance of the gradient estimates, which increases the number of iterations required to reach a given error tolerance. This type of effect is formalized in work such as Durmus et al [1] (e.g., Corollary 9 with variance $\sigma_0$ defined in Assumption MD 1 on p. 3). We have added a citation to the section on complexity. We chose to present the complexity as $\mathcal{O}(dk)$ rather than $\mathcal{O}(bdk)$ to avoid a discussion early on about batch size. We then rationalize this choice by arguing that we can simply set $b=1$ if we like (although we don’t suggest that in experiments). Technically, $b$ just needs to be constant w.r.t. $d$ and $k$ for this notation to be valid. Practically speaking, $b$ just needs to be much less than $d$, which it is in our experiments, and must be to even consider running on the high dimensional datasets we consider.
> - *Comparison with top-1 generalized eigenvector + deflation*: Deflation requires modifying the spectrum of the $A$ matrix such that any vector that is not $B$-orthogonal to the previously learned vectors achieves “zero” eigenvalue. Prompted by your feedback, we have shown our utilities can be derived from a deflation perspective and we have added it to Appx B, Proposition 4. Therefore, it is possible to take a deflation approach, but not as a direct application of “Step 1 deflate, Step 2 Gen-Oja”. To combine Gen-Oja with a deflation approach, you would have to deflate the matrix $A$ in a stochastic way by constructing unbiased estimates of the deflated $A$ matrix, $(1 - \sum_j B \frac{v_j v_j^\top}{||v_j||_B^2}) A$. Designing an algorithm that constructs unbiased estimates of gradients despite the problematic denominator terms is precisely what this work tackles. In other words, it is not clear how Gen-Oja could be used “out-of-the-box” (and probably why the Gen-Oja authors stopped at top-$1$ rather than top-$k$).
>
> [1] On Riemannian Stochastic Approximation Schemes with Fixed Step-Size. Durmus et al. AISTATS '21.

---

> > ### Author Response · Authors · 2022-11-10
> > **[U98f] Response 1b**
> >
> > - *Concurrent vs Sequential Updates*:
> > 1. As we mention above, our approach is similar in spirit to an online deflation approach where we have provided a novel way of performing the deflation despite the pesky stochastic denominator terms.
> > 2. Also, Figure 3 can give the reader some intuition for the advantages of the parallel approach over the sequential approach. 1) Intuitively, the parallel approach is similar to the sequential approach but with “warm starting”. Child eigenvectors are allowed to learn while their parents are learning, which puts them in a good position to reach their correct directions once their parents have learned. In contrast, a sequential approach would randomly initialize the child once the parents have converged, leaving the child to traverse a longer geodesic to reach its true destination. 2) How do you know when the parents are done learning? You would need to measure convergence of the eigenvectors and this is difficult in the stochastic setting. You could use a running mean of the Riemannian gradient norm or of the difference in successive Rayleigh quotients, but this is approximate. This is an interesting challenge for future research.
> > 3. We have added a figure and discussion in Appx G.2 that repeats Figure 3, but where each eigenvector completes learning once its rayleigh quotient (eigenvalue) is within 0.1 of the true value (we chose this value of 0.1 because full batch EigenGame reaches at least 0.1 accuracy for all 3 eigenvalues by the end of training). The next eigenvector then begins learning. In this experiment, the first two eigenvalues are approximated well enough, but this level of accuracy is not high enough to allow learning the 3rd eigenvector accurately. This supports what we said above where knowing when to stop learning and deflate is a difficult problem because the accuracy of parents affects the learning of children in ways that depend on the spectrum (which is clearly unknown in practice).

---

> > > ### Comment · Reviewer_U98f · 2022-11-10
> > > **Authors addressed the points I raised**
> > >
> > > The responses provided by the authors have addressed the points I raised. I have thus increased my score to an 8.

---

### Official Review · Reviewer_aVkz · 2022-10-25

**Confidence:** 4
**Correctness:** 4
**Technical Novelty And Significance:** 4
**Empirical Novelty And Significance:** 4
**Recommendation:** 8

**Clarity, Quality, Novelty And Reproducibility:**

Quality: High
Clarity: Overall good. It would be better to add the keyword "symmetric" in the title.
Originality: High

**Strength And Weaknesses:**

Strength:
1. The paper is clearly motivated and written. It is a pleasure to read.
2. The game-theoretic formulation is novel, which motivates the development of the novel algorithm.
3. The proposed algorithm is equipped with strong convergence guarantees.
4. The proposed algorithm is suitable for parallelization and can achieve O(dk) runtime complexity, which outperforms O(d^2k) in state-of-the-art algorithms.

Weakness:
1. It seems that the convergence guarantee of the stochastic version of the algorithm is missing.
2. Can this approach be applied to general non-symmetric GEP?

**Summary Of The Paper:**

This paper considers the problem of finding top-k eigenvalues of the symmetric generalized eigenvalue problem (SGEP). It proposes a game-theoretic formulation whose Nash equilibrium is the top-k eigenvalues of SGEP. Using this new result, the authors develop a parallelizable algorithm suitable for tackling SGEP with streaming data sets. It proves the asymptotic convergence of the proposed algorithm and show how to achieve O(dk) runtime complexity.

**Summary Of The Review:**

This paper proposes a game-theoretic formulation for SGEP. Based on the novel result that the Nash equilibrium of the formulation is the top-k eigenvalues of SGEP, the authors propose a novel algorithm, which shed new light on many problems that can be equivalently formulated as SGEP. The full-batch version of the proposed algorithm is equipped with theoretical guarantee. It is also applicable to streaming data set with seemingly good numerical performance.

---

> ### Author Response · Authors · 2022-11-10
> **[aVkz] Response 1**
>
> We’re glad you enjoyed reading our paper. We have added “Symmetric” to the paper tile and responded below to your questions posed under “Weaknesses”:
>
> - *“It seems that the convergence guarantee of the stochastic version of the algorithm is missing.”* That is correct. In Appx E.2, we describe the difficulties in establishing a convergence proof for the stochastic setting.
> - *“Can this approach be applied to general non-symmetric GEP?”* Unfortunately, it cannot. The difference between the SGEP and the non-symmetric GEP is non-trivial. The SGEP is guaranteed to have real eigenvectors and eigenvalues. Furthermore, the eigenvectors are guaranteed to be $B$-orthogonal. On the other hand, the non-symmetric GEP may have complex eigenvalues and eigenvectors. Also, in general, the eigenvectors will not satisfy any orthogonality properties, making it more difficult to design an algorithm for that setting. Despite these challenges, we are definitely interested in exploring ways of extending to non-symmetric settings in future work. For an example of a relevant problem, see “Quadratic eigenvalue problem: methods of solution” on wikipedia where $B$ is symmetric positive definite, but $A = \begin{bmatrix} C & K \\\ -I_n & 0 \end{bmatrix}$ is not symmetric.

---

### Decision · Program_Chairs · 2023-01-20

**Decision:**

Accept: notable-top-25%

**Justification For Why Not Higher Score:**

While interesting, the convergence results are only asymptotic. The game theoretic perspective on computing eigenvectors/eigenvalues is not entirely novel: there are other papers providing similar results but for less general problems.

**Justification For Why Not Lower Score:**

The work is interesting conceptually, there are theoretical results that justify the proposed algorithm, the algorithm is well motivated from the implementation perspective with care taken when considering modern computational architectures, and the experiments show significant improvement over the state of the art, so clearly the paper deserves to be in this pile.

**Metareview: Summary, Strengths And Weaknesses:**

The paper considers solving the symmetric generalized eigenvalue problem (SGEP), which is of the form $Av = \lambda Bv,$ where $A$ is a symmetric matrix and $B$ is symmetric positive definite. This setting generalizes PCA/SVD problems and is motivated well by learning applications such as canonical correlation analysis, independent component analysis, and computing successor features in reinforcement learning, provided in the paper. The assumptions about $A$ and $B$ are crucial in ensuring eignevalues are real and eigenvectors are $B$-orthogonal.

The paper introduces a new problem formulation for computing top $k$ eigenvectors of SGEPs corresponding to a multi-player game. The paper shows that the solutions to such games (Nash equilibria) correspond to the top $k$ eigenvectors of corresponding SGEPs. The paper then provides an algorithm for solving this problem and shows that it converges to the desired solution in an asymptotic sense. Additionally, the paper discusses different implementation issues and shows how to reduce the per iteration complexity compared to state-of-the-art by a factor scaling linearly with the problem dimension $d$. Finally, the paper provides numerical experiments that demonstrate the proposed approach is more effective than state of the art, allowing handling problem dimensions higher by a factor of a 1000 when doing an empirical analysis of neural similarity on activations.

The paper is very well written, easy to follow, and represents a well rounded piece of work. As such, it is a good fit for ICLR.

**Note From Pc:**

if the above contains the word "oral" or "spotlight" please see: "oral" presentation means -> notable-top-5% and "spotlight" means -> notable-top-25%. As stated in our emails, we are disassociating presentation type from AC recommendations

**Summary Of Ac-Reviewer Meeting:**

N/A